# Multimodal Physiological Signals Representation Learning via Multiscale Contrasting for Depression Recognition

## ABSTRACT

Depression recognition based on physiological signals such as functional near-infrared spectroscopy (fNIRS) and electroencephalogram (EEG) has made considerable progress. However, most existing studies ignore the complementarity and semantic consistency of multimodal physiological signals under the same stimulation task in complex spatio-temporal patterns. In this paper, we introduce a multimodal physiological signals representation learning framework using Siamese architecture via multiscale contrasting for depression recognition (MRLMC). First, fNIRS and EEG are transformed into different but correlated data based on a time-domain data augmentation strategy. Then, we design a spatio-temporal contrasting module to learn the representation of fNIRS and EEG through weight-sharing multiscale spatio-temporal convolution. Furthermore, to enhance the learning of semantic representation associated with stimulation tasks, a semantic consistency contrast module is proposed, aiming to maximize the semantic similarity of fNIRS and EEG. Extensive experiments on publicly available and self-collected multimodal physiological signals datasets indicate that MRLMC outperforms the state-of-the-art models. Moreover, our proposed framework is capable of transferring to multimodal time series downstream tasks. We will release the code and weights after review.

## CCS CONCEPTS

• **Computing methodologies** → **Artificial intelligence**; **Cognitive science**; • **Human-centered computing** → *HCI design and evaluation methods*.

## KEYWORDS

Depression Recognition, Multimodal Physiological Signals, Spatio-temporal Contrasting, Semantic Consistency

## 1 INTRODUCTION

Depression is a common mental disorder, which is different from regular mood changes and feelings about everyday life. Characterized by persistent feelings of sadness, lack of interest, social withdrawal, diminished social skills, and even physical symptoms such as dizziness and nausea, depression significantly affects various aspects of life, including relationships with family, friends, and the community, as well as work and study efficiency [20, 28, 49]. It

*ACM MM, 2024, Melbourne, Australia*
© 2024 Copyright held by the owner/author(s). Publication rights licensed to ACM.
ACM ISBN 978-x-xxxx-xxxx-x/YY/MM
https://doi.org/10.1145/nnnnnnn.nnnnnnn

is estimated that about 3.8% of the population are experiencing depression [33] and more than 700 thousand people die due to suicide every year [32].

The first recurrence rate of depression reaches 50% and repeated attacks significantly increase the disability rate. The significant factors affecting diagnosis and treatment are lack of resources and trained healthcare personnel. In addition, the inability to make an accurate assessment is another factor affecting effective treatment. Thus, it is urgent to enhance the accuracy of depression recognition and assessment at early stages, aiming to diminish both recurrence and disability rates. Currently, the recognition and assessment of depression mainly depend on the experienced doctors to perform clinical diagnosis based on professional scales such as the Patient Health Questionnaire (PHQ-9) [22] and Beck Depression Inventory (BDI-II) [13], as well as biomarker data. However, With the increasing number of patients, early detection is often limited and time-consuming, and subject to individual subjective observation and lack of real-time measurement. Recent strides in brain science have provided critical insights for depression diagnosis [18, 19, 36, 51, 55], with techniques like electroencephalogram (EEG) [2, 9, 31, 44] and functional near-infrared spectroscopy (fNIRS) [3, 40, 62, 63] becoming increasingly prominent due to their safety, portability, affordability, temporal precision, and minimal environmental demands. Therefore, it is necessary to explore an automatic depression recognition method based on physiological signals to assist the clinical diagnosis of doctors and accelerate the treatment for patients [15, 16, 30].

The wide collection and analysis of multimodal physiological signals such as fNIRS and EEG provide more potential to combine them to perform mental disease recognition. The distinct sampling mechanisms of fNIRS and EEG pose challenges for direct fusion at the data level, leading to a predominant focus on feature-level fusion strategies in recent research. For example, Pietro et al. employed EEG and fNIRS to classify the four symptoms of Alzheimer's disease, which achieved higher accuracy by integrating its complementary characteristics compared with single-modal experiments [5]. Shin et al. utilized typical eigenvalue scores and a common spatial pattern method to fuse the fNIRS and EEG feature [46]. Similarly, Qiu et al. proposed a multimodal feature-level fusion method, achieving good results in the classification of brain activity induced by preference music and neutral music [38]. Furthermore, Zhang et al. designed a feature fusion method based on spatio-temporal alignment strategy to obtain a significantly improved classification level in the motor imagery paradigm compared to the non-aligned method [60]. However, focusing only on feature-level fusion for EEG and fNIRS with time series property makes it easy to ignore the spatio-temporal representation and multimodal complementary features. Moreover, the existing studies have not considered the deep semantic information reflected by physiological signals under specific stimulation tasks, such as the activation status of brain regions.

To address the above issues, we propose a **M**ultimodal physiological signals **R**epresentation **L**earning framework via **M**ultiscale **C**ontrasting for depression recognition (MRLMC). This framework employs the Siamese network architecture, which utilizes two encoders with the same structure and shared weights to process different modalities. Specifically, first, fNIRS and EEG are fed into a time-domain data augmentation module to generate different but correlated data. This ensures that MRLMC learns the two types of augmented feature representation of the data. Then, we design a multiscale spatio-temporal convolution (MSC) module to learn the spatio-temporal representation and dynamic characteristics of multimodal physiological signals. The spatio-temporal contrasting module aims to minimize the differences in fNIRS and EEG feature representations while enhancing their complementary nature. Furthermore, we propose a semantic consistency module to further mine the deep semantic information such as the activation status of brain regions. It aims to maximize the semantic similarity of multimodal physiological signals. In summary, the main contributions of this paper include:

- We propose a multimodal physiological signals representation learning framework using Siamese network architecture via multiscale contrasting for depression recognition. This framework presents a novel approach to handling multimodal physiological signals and provides an objective auxiliary diagnosis.
- We design a spatio-temporal contrasting module to learn the spatio-temporal representation and dynamic characteristics. Additionally, we propose a semantic consistency module to further learn the semantic consistency representation under stimulation tasks.
- Extensive experiments are performed on publicly available and self-collected multimodal physiological signals datasets to validate the effectiveness of the MRLMC framework. The results show the superiority of the proposed method for the advancement of depression recognition.

## 2 RELATED WORK

For EEG-based depression recognition research, Rajendra et al. proposed a convolutional network for EEG data with 15 normal controls and 15 depression patients to perform depression classification and found that the signal in the right hemisphere is more active than the signal in the left hemisphere [1]. Shah et al. proposed a NeuCube model based on a pulse network to classify depression and normal controls by neural circuit connections based on EEG signals [41]. Uddin et al. captured the symptom information by combining recurrent neural networks (RNN) with long short-term memory (LSTM) [50]. Recently, Hashempour et al. proposed a hybrid convolutional and temporal-convolutional neural network to continuously estimate the BDI score to achieve depression detection [11]. Peng et al. constructed attentive simple graph convolution network and transformer neural network for depression detection and characterized the alteration of relevant neural patterns in the depressed patients [35].

For fNIRS-based depression recognition research, Liu et al. focused on stimulation tasks to investigate the advantages of fNIRS in cognitive activation and utilized the support vector machine

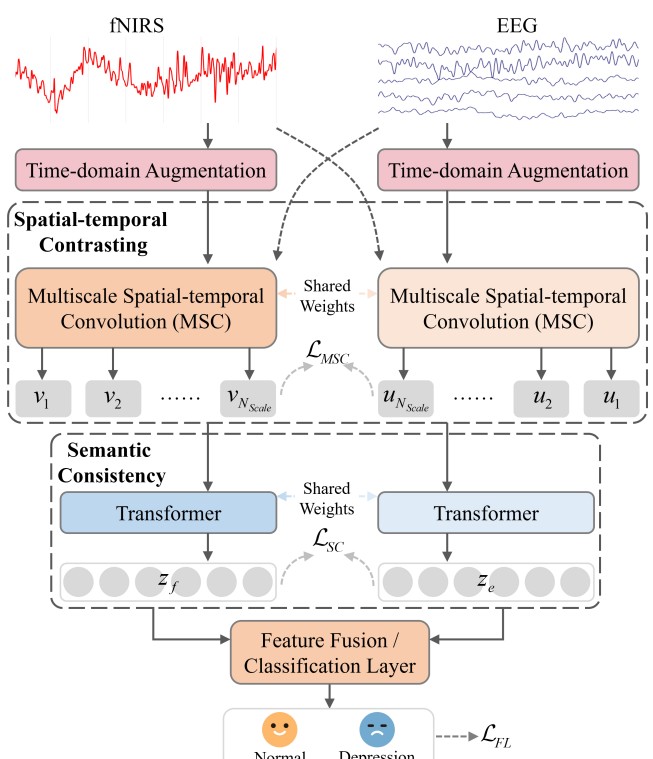

Figure 1: The overview of the MRLMC framework. The MRLMC adopts the Siamese network architecture, composed of multimodal signals input, a spatio-temporal contrasting module and a semantic consistency module.

classifier based on LSTM to perform classification tasks [26]. fNIRS data has reliably reflect cognitive profiles on the brain in different stimulation tasks [29, 39], and presents signal differences under different stimulation task time points [57]. Wang et al. proposed a transformer-based fNIRS classification network to explore spatial-level and channel-level representations of fNIRS signals to improve data utilization and feature representation [54]. Similarly, Zhang et al. achieved mild cognitive impairment recognition by exploiting the multidimensional features of fNIRS data including channel, temporal, and spatial features [59]. Wang et al. transformed fNIRS signals into 2-D wavelet feature maps by using wavelet transform and parallel-CNN feature fusion to diagnose depressive disorder [52]. However, these works mentioned above ignore the nonlinear and segment characteristics of EEG and fNIRS. In addition, ignoring the dynamic characteristics and semantic representation of neural activity under stimulation tasks results in weak classification performance.

There are many brain-computer studies on multimodal recognition tasks based on fNIRS and EEG but less research in the area of multimodal depression recognition. He et al. proposed a multimodal multitask neural network model to fuse the EEG and fNIRS signals to achieve motor imagery classification [14]. Gao et al. utilized an EEG-informed fNIRS general linear model to extract common spatial pattern features and the support vector machine was used as

Figure 2: The input modes of multimodal signals in MRLMC, including single modal mode and multimodal mode.

the classifier [8]. Differently, we establish a multimodal contrastive learning framework based on the Siamese network architecture. fNIRS and EEG are fed into the spatio-temporal contrasting module and semantic consistency module to extract complementary features, dynamic features and semantic consistency representations to realize multimodal depression recognition.

## 3 METHODOLOGY

In this section, we describe the components of MRLMC framework in detail. As shown in Figure 1, the MRLMC framework adopts the Siamese network architecture to learn the feature representations of fNIRS and EEG signals. Specifically, we first utilize the time-domain data augmentation method to generate different but correlated data. Then, we design a spatio-temporal contrasting module to extract the feature representation and dynamic characteristics of the physiological signals. Finally, a deep semantic representation of fNIRS and EEG signals is achieved through the semantic consistency module. This multimodal semantic representation is then fused and fed into the classification layer to realize depression recognition.

### 3.1 Multimodal Signals Input Modes

The collection of fNIRS and EEG data involves stringent conditions, which present challenges due to limited medical resources and the prevalent stigma associated with patients. Therefore, in scenarios with limited data, the data augmentation method plays an important role, and it is also a key part of realizing single-modal contrasting learning. As shown in Figure 2, when only singlemodal (either fNIRS or EEG) is available, both the raw and augmented data are utilized as pairs. When the input is fNIRS and EEG, they are shaped as a pair of data, with the data augmentation strategy randomly applied to part of the data. The commonly used jitter-and-scale strategy and permutation-and-jitter strategy data augmentation methods do not consider both the collection paradigm and the process of physiological data. Since the physiological data for depression patients are mostly collected with specific stimulation tasks, the time-domain augmentation methods including time

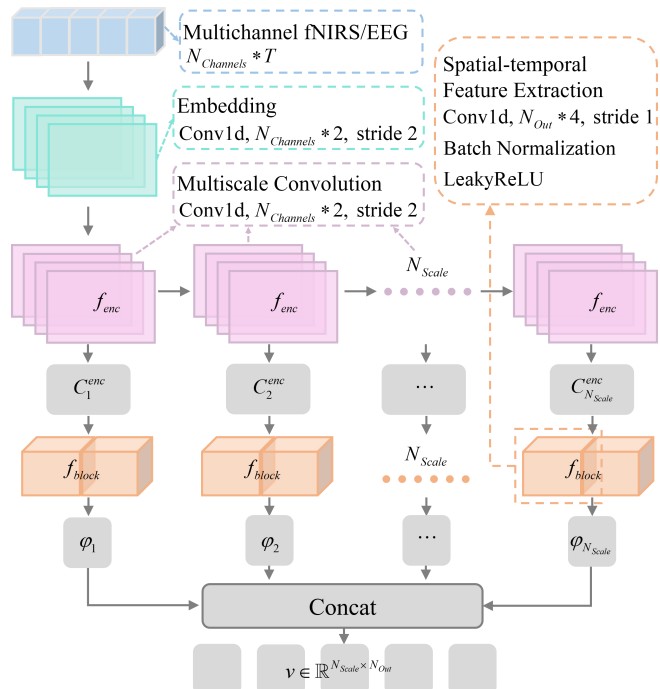

Figure 3: The overview of multiscale spatio-temporal convolutional (MSC) network. The input raw data or augmented data undergoes a convolution layer to generate embedding. Then, the spatio-temporal representation is extracted by multiscale convolution.

warping and time masking [42] are utilized to generate different but correlated data.

Given a sample $x$, the time step of the time masking method is $[t_0, t_0 + t_{tm}]$, where $t_0 \in [0, t_q)$, and the masking parameter $t_{tm} \in (0, \lambda], \lambda \le t_q$, introducing an upper bound that the width of the time masking cannot be larger than the response time of each question of stimulation task. Similarly, the time step of the time warping method is $[t_0, t_0 + t_{tw}]$, where $t_0 \in [0, t_q)$ and the warping parameter $t_{tw} \in (0, \lambda], \lambda \le t_q$. The augmented data is denoted as $x'$, which has the same time scale as $x$. Formally, let $D_{Multi} = \{F_i, E_i\}^N$ be a dataset of fNIRS and EEG, such that each fNIRS sample $F_i$ corresponds to EEG sample $E_i$. For each input sample $F_i$ and $E_i$, we denote the augmented data as $x_i$ and $y_i$, $D = \{x_i, y_i\}^N$ as the input data. In the case of single modal inputs, $x_i$ and $y_i$ denote raw data and augmented data respectively and the $N$ is the number of samples. Multimodal data is fed into the spatio-temporal contrasting module to extract latent representation.

### 3.2 Spatio-temporal Contrasting

Physiological signals, as a kind of multichannel time series data, are characterized by spatio-temporal features that are the most important kind of representation. Specific stimulation tasks are usually performed to collect physiological signals. When the participants are handling stimulation tasks, the status of the brain is transformed from a resting state to an activated state. Regarding

the time dimension, physiological signals have dynamic changing characteristics. Meanwhile, the prefrontal areas of the individual brain are associated with emotional expression, and different channels have similar but different characteristics. Therefore, we design a spatio-temporal contrasting module, as shown in Figure 1, which utilizes the contrastive loss to minimize the differences between fNIRS and EEG feature representations and maximize complementarity through extracting the spatio-temporal representations of raw data and augmented data. Figure 3 presents the MSC network, which extracts the spatio-temporal representation and dynamic characteristics of physiological signals.

Given an input signal $x$, its dimension is $N_{Channel} \times T$, where $N_{Channel}$ is the number of channels of data and $T$ is the collection duration, which is determined by the collection device and the data type. Then, the $x$ is fed into the encoder to get the latent representation. The encoder based on the convolution layer maps $x$ into a latent representation $C = f_{enc}(x)$, $C \in \mathbb{R}^d$, where $d$ is the dimension of the feature. Thus, we get $C$ for the feature representation of a physiological signal, which is then fed into the multiscale convolution layers. The $C$ is passed to the $N_{Scale}$ layer multiscale convolution to extract high-dimensional representations $C^{enc}$. Then, the representations are fed into $N_{Scale}$ spatio-temporal feature extraction blocks $f_{block}(\cdot)$ to extract spatio-temporal representation of physiological signals. Finally, we get spatio-temporal representation $v$ of a physiological signal,

$$v = Concat(\varphi_1, \varphi_2, \cdots, \varphi_{N_{Scale}}), \quad (1)$$

where

$$\varphi_i = max(\alpha * Norm(f_{block}(C_i^{enc})), Norm(f_{block}(C_i^{enc}))), \quad (2)$$

which simplified to $v = [\varphi_1, \varphi_2, \cdots, \varphi_{N_{Scale}}]$, $v \in \mathbb{R}^m$, where $m = N_{Scale} \times N_{Out}$ is the dimension of feature, $N_{Out}$ is the output dimension of the spatio-temporal feature extraction blocks and $\alpha$ is the control weight.

Through the spatio-temporal contrasting module, the multimodal data generate spatio-temporal representations $v$ and $u$, where $u$ is generated from another modal or augmented data. Given a batch of input samples denoted as $N = batch\_size$, we get $2N$ items from fNIRS and EEG. For a $u$ item, we denote $u^+$ as the positive sample for $v$, and thus $(v, u^+)$ are considered as the positive pair. The other $(2N - 2)$ items in the same batch are considered negative samples for $v$, then $v$ forms negative pairs with $(2N - 2)$ negative samples. Therefore, we can define the spatio-temporal contrasting loss to maximize the similarity between positive pairs and the difference between negative pairs.

Given the $v$ and $u$ items, we compare the similarity of positive pair $(v, n^+)$ with the similarity of $(2N-2)$ negative pairs, the spatio-temporal contrasting loss $\mathcal{L}_{MSC}$ is defined as follows:

$$\mathcal{L}_{MSC} = -\log \frac{exp(sim(v, u^+)/\tau)}{exp(sim(v, u^+)/\tau) + \sum_{j=1}^{2N-2} exp(sim(v, u_j)/\tau)}, \quad (3)$$

where $sim(\cdot)$ denotes cosine similarity,

$$sim(v, u) = \frac{v^T u}{\|v\| \|u\|}, \quad (4)$$

where $\tau$ is a temperature parameter. Through the spatio-temporal contrasting loss $\mathcal{L}_{MSC}$, the differences of feature representations

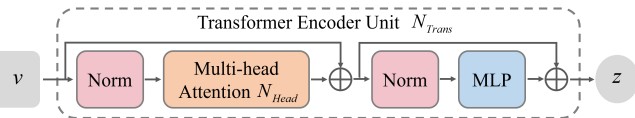

**Figure 4: The architecture of transformer unit in semantic consistency module.**

between fNIRS and EEG could be minimized, which also maximizes the complementarity of these two representations. And then the spatio-temporal representations are fed into the semantic consistency module to further learn deep semantic information.

## 3.3 Semantic Consistency

The depression patients are characterized by persistent low mood, pleasure deficit, and cognitive impairment, which presents the difference with the control group on the brain activity level and activation state when performing the stimulation task [42]. fNIRS and EEG reflect brain activation state by detecting slight changes in brain activity, so it is necessary to mine deeper semantic information that can reflect brain activation state. We propose a semantic consistency module to maximize the semantic similarity of multimodal physiological signals and further mine deep semantic information such as brain activation state.

We utilize the transformer unit as the semantic feature extraction model because of its context-awareness. The architecture of the transformer unit is shown in Figure 4, which mainly consists of successive blocks of multi-head attention (MHAttn) and MLP. The MLP block consists of two fully connected layers and a non-linear ReLU. The transformer unit is defined by the following equations:

$$MHAttn(Q, K, V) = Concat(Head_1, Head_2, \cdots, Head_{N_{Head}})\mathcal{W}^O \quad (5)$$

where $Q$ represents the input feature vector, $K$ represents the key vector, $V$ denotes the value vector, $N_{Head}$ represents the number of heads, and $\mathcal{W}^O$ denotes the final output weights. $Head_i$ is defined as follows:

$$Head_i = Attention(Q\mathcal{W}_i^Q, K\mathcal{W}_i^K, V\mathcal{W}_i^V) \quad (6)$$

where $\mathcal{W}_i^Q, \mathcal{W}_i^K, \mathcal{W}_i^V$ denote the weight matrics of $Q, K, V$, respectively. $Attention(\cdot)$ is define as

$$Attention(Q, K, V) = softmax(\frac{QK^T}{\sqrt{d_K}})V \quad (7)$$

where $d_K$ denotes the dimentional size of vector $K$. Given the spatio-temporal representations $v$, we pass it through the transformer unit as follows:

$$\psi_i = MHAttn(Norm(v_{i-1})) + \psi_{i-1}, 1 \le i \le N_{Trans}, \quad (8)$$

and then the $\psi_i$ is input to the MLP block:

$$z_i = MLP(Norm(\psi_i)) + \psi_i, 1 \le i \le N_{Trans}, \quad (9)$$

where $N_{Trans}$ denotes the number layers stacked to generate the final feature $z$.

Given the multimodal spatio-temporal representations $v$ and $u$, a multilayer stacked transformer unit is utilized to extract the semantic feature $z^f$ and $z^e$. The dimension size of $z^f$ and $z^e$ are

the same as $v$ and $u$. We utilize cosine similarity as the semantic consistency loss to maximize the semantic similarity of multimodal physiological signals. The semantic consistency loss can be denoted as follows:

$$\mathcal{L}_{SC} = sim(z^f, z^e). \tag{10}$$

## 3.4 Depression Recognition

Ultimately, $z^f$ and $z^e$ are concatenated and fed into the classification layer for depression recognition, which includes two fully connected layers and the ReLU layer. In real healthcare scenarios, the collected dataset exists the class imbalance problem, so the focal loss function is utilized to perform depression recognition, which is defined as follows:

$$\mathcal{L}_{FL} = -\alpha(1-P)^{\gamma} log(P), \tag{11}$$

where $P$ denotes the predictive probability of the model, $\alpha$ is the weighting factor to balance the positive and negative samples, and $\gamma$ is the adjustable parameter. The adjustment factor $(1-P)^{\gamma}$ can be adjusted adaptively according to the difficulty of the sample. In instances where samples are inherently easier to classify, the parameter $P$ is larger, causing the adjustment factor to tend to zero. Consequently, this results in a reduced impact on the loss function, prompting the model to focus more on samples that are difficult to classify. The overall loss is the combination of the spatio-temporal contrasting loss, semantic consistency loss, and classification loss as follows:

$$\mathcal{L} = \lambda_1 \mathcal{L}_{MSC} + \lambda_2 \mathcal{L}_{SC} + \mathcal{L}_{FL}, \tag{12}$$

where $\lambda_1$ and $\lambda_2$ are fixed scalar hyperparameters denoting the relative weight of each loss.

## 4 EXPERIMENTS

The datasets and implementation details are first presented in this section. We then conducted extensive experiments to validate the effectiveness of the MRLMC framework.

## 4.1 Datasets Description

To evaluate the performance of our proposed method, we conduct a series of experiments on two datasets.

**MODMA** dataset [25] is a publicly available dataset, and we only use event-related EEG data, including 53 participants (24 outpatients diagnosed with depression and 29 healthy controls). It uses a Dot-probe stimulation task to record EEG signals. The Dot-probe is composed of facial pictures from the standardized native Chinese Facial Affective Picture System [27]. The facial pictures are classified into four sets as fear, sad, happy, and neutral emotions based on their valence. Any two facial images of different valences appear on the screen. During the experiment, participants were asked to focus on the screen and watch freely with their eyes. When the dot appeared, they were asked to press the button quickly and accurately without making any body movements, including head or legs, and as much as possible without making unnecessary eye movements, glances and blinks. Continuous EEG signals were recorded using a 128-channel device. The sampling frequency was 250 Hz.

**fNIRS-EEG** dataset is a self-collected multimodal physiological signals dataset, including fNIRS and EEG signals. We utilize a verbal fluency stimulation task to record data, including 96 participants

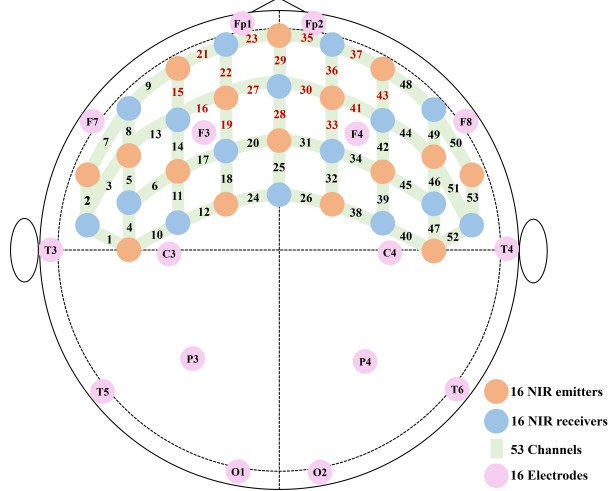

**Figure 5: The channel location of fNIRS and EEG. Among them, orange is 16 NIR emitters, blue is 16 NIR receivers, green is 53 fNIRS channels, and purple is 16 EEG channels.**

(79 depression patients and 17 healthy controls) for only fNIRS, and 64 participants (52 depression patients and 12 healthy controls) for both fNIRS and EEG. During the data collection process, doctors helped participants wear the device to ensure the probe was tightly attached to the scalp until the channel pass rate reached 80%. The entire stimulation task includes a pre-task silence period, a task period and a post-task silence period. The silent period required participants to sit up straight in front of the computer, remain calm, and not shake their bodies. During the task period, three questions appear on the computer screen, and participants are asked to name the fruits, appliances and vegetables they can associate with the questions. As shown in Figure 5, the near-infrared device used in this study has 16 near-infrared (NIR) emission probes and receiving probes, and a total of 53 channels are connected. The detector emits near-infrared light at 690nm and 830nm. Throughout the test period, the NIR device collected the intensity of the emitted light at two wavelengths at a sampling frequency of 100hz. Through the test, each participant had data of 150×100×53×2, where 150 is the duration of the test, 100 is the data collection frequency, 53 is the number of channels and 2 is the number of wavelengths. The EEG device used in this study has 16 channels, and the electrode-wearing method follows the 10-20 lead system standard. The EEG device collects electrical signals at a sampling frequency of 1000hz, with data for each participant of 150×1000×16, where 150 is the duration of the test, 1000 is the data collection frequency, 16 is the number of channels.

## 4.2 Implementation Details

*4.2.1 Experimental Setup.* The entire dataset is randomly split into training set, testing set and validation set for each training phase. The final modal used in testing is the one that exhibits the best performance on the validation set. For the evaluation of depression diagnosis, the macro *Accuracy*, *Precision*, *Recall* and *F1-score* are used as evaluation indicators for the performance of the model.

**Table 1: Model configuration parameters.**

| Parameters | Values |
|---|---|
| Learning rate | $1e-3$ |
| Batch size | 16 |
| Dropout | 0.1 |
| $N_{Scale}$ | 5 |
| $N_{Trans}$ | 1 |
| $N_{Head}$ | 16 |

Multiple experiments were conducted to take the average value of the evaluation indicators.

*4.2.2 Data Preprocessing.* For the MODMA dataset, we utilized the EEGLAB toolkit [6] within the MATLAB platform for EEG denoising. We applied a bandpass filter with a frequency range of 1-40 Hz to the raw EEG signal. Then, we utilized the extended ICA algorithm to obtain multiple independent EEG components to eliminate artifacts and noise components, such as electrooculogram (EOG), ECG, EMG, and eye movement. Finally, the ICLabel plugin was used to remove the identified artifacts and noise components. Additionally, we only selected part of the channel data from the prefrontal brain area, which processes emotional expression.

For the fNIRS-EEG dataset, we first utilized the near-infrared data analysis tools for fNIRS data preprocessing. The preprocessing steps begin with the elimination of motion artifacts unrelated to the raw data using the temporal derivative distribution repair method. Subsequently, the light intensity signal was converted into an optical density profile, which was then filtered using the finite impulse response band-pass filter with 0.01-0.08Hz to eliminate noise caused by physiological fluctuations such as pulse and respiration and baseline drift caused by environmental and temperature changes. Finally, the optical density data were converted to concentration change of oxygenated hemoglobin (HbO) and deoxy-hemoglobin (HbR) using a modified Beer-Lambert method. Based on fNIRS-based research [3, 10, 63], this study also deliberately focused on the HbO concentration change data in subsequent method design. Additionally, we only selected part of the channels, which are the red font channels shown in Figure 5. For the EEG signals, the same preprocessing method as the MODMA dataset was implemented. Specifically, resampling was implemented for both fNIRS and EEG data.

*4.2.3 Model Configuration.* The model is constructed using the Pytorch framework and optimized using the RMSprop optimizer. The learning rate, batch size and other parameters are shown in Table 6.

## 4.3 Experimental Results

*4.3.1 EEG Depression Recognition.* To demonstrate the effectiveness of the MRLMC model and its applicability on single modal modes, we first conducted sufficient experiments on the MODMA dataset. The benchmark algorithms include EEGNet [23], STGCN [56], DGCNN [48], HGP-SL [61], SAGE [24], SST-Emotionnet [17], SGP-SL [4], CGIPool [34], SGP-SL [4], TSception [7], CLG [43], dFL [45] and 1DEEG-Transformer [37] for comparison. All models utilize

**Table 2: Comparison of MRLMC model with baseline methods on MODMA dataset.**

| Model | Acc. | Prec. | Rec. | F1. |
|---|---|---|---|---|
| EEGNet [23] | 0.568 | - | 0.668 | 0.600 |
| STGCN [56] | 0.588 | - | 0.577 | 0.596 |
| DGCNN [48] | 0.597 | - | 0.459 | 0.552 |
| HGP-SL [61] | 0.585 | 0.536 | 0.625 | 0.577 |
| SAGE [24] | 0.679 | 0.640 | 0.667 | 0.653 |
| SST-Emotionnet [17] | 0.736 | 0.692 | 0.750 | 0.720 |
| CGIPool [34] | 0.736 | 0.692 | 0.750 | 0.720 |
| SGP-SL [4] | 0.849 | 0.808 | **0.875** | 0.840 |
| TSception [7] | 0.544 | - | 0.445 | 0.486 |
| CLG [43] | 0.765 | - | 0.757 | 0.759 |
| dFL [45] | 0.750 | - | 0.614 | - |
| 1DEEG-Transformer [37] | 0.782 | 0.784 | 0.692 | 0.749 |
| MRLMC | **0.867** | **0.875** | **0.875** | **0.864** |

the raw EEG signals. Table 2 exhibits the evaluation indicators for each model. For EEG-based depression recognition, the MRLMC model attains the most superior performance with 0.867, 0.875, 0.875, and 0.864 in accuracy, precision, recall, and F1-score, respectively. Specifically, the highest recognition accuracy 0.867 was obtained by MRLMC. EEGNet is the most classic convolutional neural network for processing EEG signals, which uses temporal and spatial convolution to extract data features. The CLG and 1DEEG-Transformer stack back and forth the convolutional layers and long short term memory network to extract temporal and spatial features. Differently, the MRLMC model designs an MSC network to extract the spatio-temporal representation and learns effective feature based on the contrastive loss function, thereby achieving the most advanced classification performance. Compared with SGP-SL, the recognition accuracy of the MRLMC model is improved by 2%. With the latest research such as the CLG and 1DEEG-Transformer, the recognition accuracy is improved by 11%. In addition, the DGCNN and CGIPool models construct the extracted features into a graph structure and mine the relationships between the channels of data. Based on existing research, it has been shown that the prefrontal lobe area of the brain performs emotional expression, which is gradually activated when a stimulation task is performed. Therefore, we implemented a channel selection process before feature extraction. Compared to the DGCNN and CGIPool networks, the MRLMC model improves by 18% in accuracy since the proposed MSC module can also extract channel features. Especially, we also mine the deep semantic information of the data, aiming to mine semantic features such as brain activation levels, and maximize the semantic representation of multimodal data based on consistency loss.

*4.3.2 fNIRS Depression Recognition.* Table 3 shows the performance of the MRLMC model on fNIRS data in the fNIRS-EEG dataset. To evaluate the superiority of our method, the baseline methods selected are Logistic Regression (LR), K-Nearest Neighbor (KNN), Support Vector Machine (SVM) [47], AlexNet [21], Residual

**Table 3: Comparison of MRLMC model with baseline methods on fNIRS-EEG dataset (only fNIRS).**

| Model | Acc. | Prec. | Rec. | F1. |
|---|---|---|---|---|
| LR | 0.813 | 0.300 | 0.583 | 0.355 |
| KNN | 0.729 | 0.188 | 0.219 | 0.188 |
| SVM [47] | 0.823 | 0.000 | 0.000 | 0.000 |
| AlexNet [21] | 0.830 | 0.790 | 0.830 | 0.800 |
| ResNet [12] | 0.720 | 0.670 | 0.720 | 0.700 |
| RF [63] | 0.833 | 0.625 | 0.175 | 0.267 |
| XGB [63] | 0.833 | 0.525 | 0.413 | 0.446 |
| Corr-AlexNet [53] | 0.900 | **0.910** | 0.900 | **0.880** |
| GCN [58] | 0.854 | 0.700 | 0.488 | 0.563 |
| Diffpool [58] | 0.875 | 0.750 | 0.475 | 0.571 |
| MRLMC | **0.913** | 0.827 | **0.908** | 0.834 |

**Table 4: Extensive experiments of MRLMC model on fNIRS-EEG dataset.**

| fNIRS | EEG | Aug. | Acc. | Prec. | Rec. | F1. |
|---|---|---|---|---|---|---|
| ✓ | ✗ | ✓ | 0.907 | 0.816 | 0.839 | 0.802 |
| ✗ | ✓ | ✓ | 0.875 | 0.834 | 0.822 | 0.771 |
| ✓ | ✓ | ✗ | 0.907 | 0.836 | 0.875 | 0.816 |
| ✓ | ✓ | ✓ | **0.917** | **0.850** | **0.881** | **0.831** |

Network (ResNet) [12], Random Forest (RF) [63], XGB [63], Corr-AlexNet [53], GCN [58] and Diffpool [58]. Our proposed method achieved 0.913, 0.827, 0.908 and 0.834 in accuracy, precision, recall and F1-score, respectively, which are satisfactory results. The accuracy of traditional machine learning methods such as LR, KNN and SVM is not satisfactory, while the accuracy of deep learning algorithms such as AlexNet is relatively improved, which highlights the superior performance of deep learning algorithms in depression recognition based on physiological signals. The Corr-AlexNet, GCN and Diffpool networks compared to traditional machine learning improve the accuracy by about 8%. These methods rely on manually extracted features for learning and lack deep exploration of spatio-temporal representation, dynamic features, and semantic representation. The MRLMC model extracts the spatio-temporal representation and dynamic features of the data through the spatio-temporal contrasting module. Additionally, the main symptoms of patients with depression include low mood and slow thinking, which causes their brains to be activated differently when performing stimulating tasks. The MRLMC model utilizes the semantic consistency module to dig deep into the semantic representation to reflect brain activation states. Compared with traditional machine learning, the accuracy is improved by about 11%, and compared with the method of manually extracting features for recognition, the accuracy is improved by about 1.5%. Overall, based on task-state physiological data, extracting spatio-temporal representation and semantic representation can achieve higher recognition accuracy.

*4.3.3 Multimodal Depression Recognition.* Table 4 exhibits the recognition results of the MRLMC model on the fNIRS-EEG dataset. The

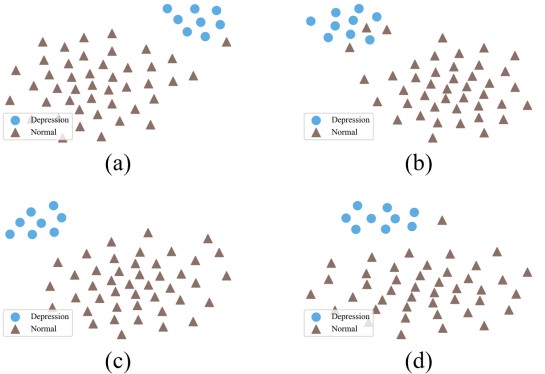

(a)      (b)

(c)      (d)

**Figure 6: The visualization of the distribution of features extracted by each module of the proposed model. (a) and (b) are the representations of fNIRS and EEG extracted by the spatio-temporal contrasting module. (c) and (d) are the semantic features of fNIRS and EEG extracted by the semantic consistency module.**

excellent results were achieved based on both fNIRS and EEG, with accuracy, precision, recall and F1-score reaching 0.917, 0.850, 0.881 and 0.831, respectively. When only based on fNIRS or EEG, the recognition accuracy reaches 0.907 and 0.875 respectively. Relying on single modal physiological signal for depression recognition, the recognition accuracy is limited by the available feature representations of data. When utilizing multimodal physiological signals, the classification performance improves by 3%. When continuing to perform the data augmentation method, the evaluation indicators all improved. fNIRS collects HbO concentration change data and EEG is an electric signal, and there are complementary features between them. The MRLMC model utilizes spatio-temporal contrasting module to learn the complementary feature representations of multimodal data. Subsequently, the proposed semantic consistency module extracts the semantic features of multimodal physiological signals, such as the degree of brain activation, which are jointly learned through consistency loss. Considering the challenge of class imbalance in real diagnosis and treatment environments, we use the focal loss function to construct a classification network, which enhances the robustness of the network to achieve higher recognition accuracy. The MRLMC model proved effective even for small-scale datasets.

To intuitively demonstrate the effectiveness and feature representation capabilities of the various modules in the MRLMC model, Figure 6 displays the distribution of features extracted by each module on fNIRS-EEG dataset. Figure 6 (a) and (b) demonstrate the distribution of representations of fNIRS and EEG extracted by the spatio-temporal contrasting module, albeit not completely separable. Figure 6 (a) and (b) represent the distribution of semantic features extracted by the semantic consistency module, at which point the MRLMC model can accomplish depression recognition.

*4.3.4 Ablation Analysis.* To verify the effectiveness of different modules in our proposed model, we conduct additional ablation experiments on the fNIRS-EEG dataset, as shown in Table 5. $\mathcal{L}_{MSC}$ and $\mathcal{L}_{SC}$ are the loss functions applied by the spatio-temporal

**Table 5: Results of loss terms ablation experiments in each proposed module.**

| $\mathcal{L}_{MSC}$ | $\mathcal{L}_{SC}$ | $\mathcal{L}_{FL}$ | Acc. | Prec. | Rec. | F1. |
|---|---|---|---|---|---|---|
| × | × | ✓ | 0.800 | 0.527 | 0.543 | 0.533 |
| ✓ | × | ✓ | 0.891 | 0.777 | 0.723 | 0.740 |
| × | ✓ | ✓ | 0.875 | 0.770 | 0.714 | 0.723 |
| ✓ | ✓ | ✓ | **0.917** | **0.850** | **0.881** | **0.831** |

**Table 6: Performance of MRLMC model with different parameters on fNIRS-EEG dataset.**

| $N_{Scale}$ | $N_{Trans}$ | $N_{Head}$ | Acc. | Prec. | Rec. | F1. |
|---|---|---|---|---|---|---|
| 4 | 1 | 16 | 0.907 | 0.815 | 0.839 | 0.806 |
| **5** | **1** | **16** | **0.917** | **0.850** | **0.881** | **0.831** |
| 6 | 1 | 16 | 0.917 | 0.838 | 0.809 | 0.804 |
| 5 | 2 | 16 | 0.891 | 0.786 | 0.777 | 0.764 |
| 5 | 3 | 16 | 0.875 | 0.530 | 0.571 | 0.549 |
| 5 | 1 | 4 | 0.792 | 0.661 | 0.738 | 0.657 |
| 5 | 1 | 8 | 0.900 | 0.795 | 0.814 | 0.787 |
| 5 | 1 | 32 | 0.896 | 0.781 | 0.798 | 0.775 |

contrasting and semantic consistency modules respectively. $\mathcal{L}_{FL}$ is the depression recognition loss, which is utilized in all experiments. The results indicate that satisfactory performance is obtained when utilizing all losses. The performance of using $\mathcal{L}_{MSC}$ or $\mathcal{L}_{SC}$ alone is better than using only recognition loss. This proves that our proposed $\mathcal{L}_{MSC}$ and $\mathcal{L}_{SC}$ can help the model obtain useful spatio-temporal representation and semantic information. This means that the spatio-temporal contrasting and semantic consistency modules are effective for multi-modal physiological signals for depression recognition.

*4.3.5 Parameter Analysis.* To further investigate the MRLMC model, we analyze in detail the impact of several important parameters of the model on performance in this section. Table 6 exhibits the performance of the MRLMC model with different parameters on the fNIRS-EEG dataset. The first three rows of indicators verify the effects of the number of spatio-temporal convolution blocks on model performance, the middle two rows verify the effects of the number of transformer units, and the last two rows verify the effects of multi-head attention. The results show that different parameters have different effects on the model. When the number of convolution blocks is 5 or 6, the recognition accuracy reaches excellent results. The number of transformer encoder units has a slightly greater impact on the performance of depression recognition. As the number of units increases, the network complexity increases, which causes overfitting of the model. In addition, information may be lost or confused during the transmission process, making it difficult for the network to learn useful semantic information. When the number of multihead attention is 8 or 16, the model achieves superior performance.

Figure 7 shows the performance of the MRLMC model with different number of spatio-temporal convolution blocks on the fNIRS-EEG dataset. As the number of convolution blocks increases, the

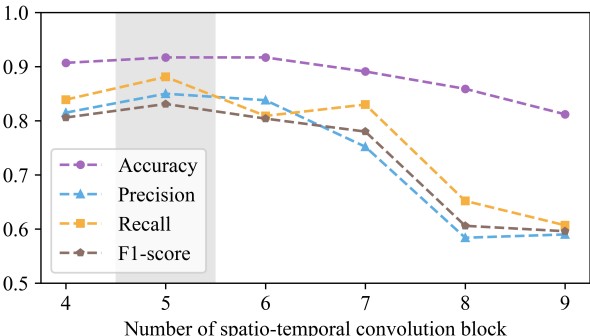

**Figure 7: Performance of MRLMC model with the different number of spatio-temporal convolution block on fNIRS-EEG dataset. The shadow part represents the superior performance.**

recognition accuracy decreases, which proves that spatiotemporal representation has a great impact on the recognition performance for small-scale datasets. The increased number of blocks means that the complexity of the model increases. The main characteristics of multimodal physiological signals are their spatio-temporal representation and dynamic variability, and their key information is often hidden in local sequence patterns and global temporal dependence. Networks with high complexity may fail to capture these key information, making it difficult to learn effective spatio-temporal representation. Therefore, for the small-scale fNIRS-EEG dataset, the results of spatio-temporal convolution blocks of 5 or 6 are most excellent. If the MRLMC model is to be transferred to other downstream tasks of multimodal time series, the number of spatio-temporal convolution blocks needs to be determined based on the characteristics of the data.

## 5 CONCLUSION

In this paper, we propose a multimodal physiological signals representation learning framework via multiscale contrasting for depression recognition. The Siamese network architecture is utilized to maximize the complementarity between multimodal data. We design multiscale spatio-temporal convolution to obtain more discriminative spatio-temporal representations and dynamic features. The spatio-temporal contrasting module aims to minimize the feature representation and maximize the complementarity of fNIRS and EEG. Meanwhile, the semantic consistency module captures contextual information and the deep semantic information of the data to maximize the semantic representation of multimodal data based on semantic consistency loss. Extensive experiments are implemented on MODMA and fNIRS-EEG datasets, and our proposed model achieves state-of-the-art performance on both singlemodal and multimodal data. Moreover, the analysis of the feature distribution and key parameters of each module shows that each module plays an important role in mining spatio-temporal representations and semantic features. Notably, the proposed model is a generalized architecture based on multichannel physiological signals, which can be extended to other mental disorders and cognitive ability recognition in the future.

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
