# OpenReview forum: "Multimodal Physiological Signals Representation Learning via Multiscale Contrasting for Depression Recognition"
_acmmm.org/ACMMM/2024/Conference — MM2024 Poster_

### Official Review · Reviewer_pwJk · 2024-05-20

**Rating:** 3
**Confidence:** 3

**Summary:**

The research presents a multimodal physiological signals framework, MRLMC, for depression recognition. It enhances the complementarity and semantic consistency of fNIRS and EEG data during identical stimulation tasks, overcoming the limitations of previous approaches. By applying time-domain data augmentation, the framework generates diverse but related datasets. A spatio-temporal contrasting module uses multiscale convolution to learn representations, while a semantic consistency contrast module maximizes the similarity in meaning between fNIRS and EEG signals. Experiments show MRLMC surpasses current models in depression detection accuracy and is extendable to other multimodal time series applications.

**Strengths:**

1. The research issue is meaningful.
2. The writing and story-telling is good.
3. Figures are clear and ablation study is sufficient.

**Limitations:**

1. In Table 3 and Table 4, why are the results of MRLMC in Table 3 not the same as those of the first line in Table 4? I think these two are both results with single fNIRS only, and they should be the same.
2. I am a bit confused about the semantic consistency loss (equation 10), which is aimed at maximizing the cosine similarity between fNIRS and EEG representations. Shoud it be 1-sim(z^f,z^e)? Moreover, it is not very intuitive to simply push the two modalities togather as this may lead to the loss of complementary information. Although the ablation study in Table 5 seems to
3. In  Section 4.2.1, the authors claimed that they conducted mulitple experiments and reported the average value. I think standard deviation should also be reproted to show the stability of each method.
4. How did the training, testing, and validation set splitted? Did the training, testing, and validation set contain samples from the same subject? Each set should exclude subjects from each other to obtain a convincing result.

**Suitability:**

3

---

### Official Review · Reviewer_Wrku · 2024-05-21

**Rating:** 3
**Confidence:** 4

**Summary:**

This paper proposes a multi-modal framework called MRLMC, which combines fNIRS and EEG signals to realize depression prediction. The authors design a spatio-temporal contrasting module and a semantic consistency module to get the feature vector from the two input signals. At last, the features are integrated to predict the depression status.

**Strengths:**

1. Exhaustive experiments on both public and self-collected datasets were conducted to prove the effectiveness of the proposed MRLMC, and the results showed that this method was superior to other methods with both single-modal and multi-modal inputs.
2. It is innovative to combine two kinds of physiological signals, EEG and fNIRS, to achieve better prediction results for depression.

**Limitations:**

1. How did the proposed MSC module extract “spatial-temporal representations” as it only consists of several common convolution layers? It seems that only because the input data contains spatial-temporal information that the module can get the result.
2. In section 4.3.2, some statements like “Compared with traditional machine learning, the accuracy is improved by about 11%”. is not precise. How did the authors get the number? Based on the average of the results of traditional machine learning methods or the result of one of the methods?
3. A concern is that MRLMC seems to be the combination of several convolution layers and transformer blocks, so the novelty of the proposed method is insufficient.

**Suitability:**

3

---

### Official Review · Reviewer_6fvE · 2024-06-07

**Rating:** 6
**Confidence:** 3

**Summary:**

This paper introduces a multimodal physiological signals learning framework, using a siamese architecture for depression recognition. This uses near-infrared spectroscopy (fNIRS) and EEG. Extensive evaluation is performed on publicly available datasets and comparisons performed with existing state of the art. Results show that this framework outperforms the state of the art.

**Strengths:**

This paper adopts a novel approach with the use of siamese architecture. Extensive evaluation is performed showed very promising results which outperform the current SOTA. Paper is very well written. Strong analysis of results is also performed.

**Limitations:**

Tables 4, 5, and 6 could be better explained in the text and are a little difficult to follow.
Please correct the following typos:
line 80 - lower case 'w'
208/9 - Clean up / re-word this sentence, it is a little difficult to follow.
447- - define(d)
475 - the collected dataset exists the class imbalance problem - Please re-word this sentence, it is a little difficult to follow

**Suitability:**

3

---

### Meta-Review · Area_Chair_SsCE · 2024-07-10

**Recommendation:** Accept (Poster)
**Confidence:** 5

**Metareview:**

Final decision - accept

Congrats! All reviewers saw merit in the work

Please ensure that all comments from the reviewers are addressed as per the rebuttal. In addition, the following discussion arose from the reviewers:
-A clearer explanation of how the convolution layer is used to extract temporal features from the input data.
-According to Table 3, the proposed MRLMC brought about 9% improvement compared to SVM. How did the authors get the number "11%"?